# A Hybrid Workflow of Residual Convolutional Transformer Encoder for Breast Cancer Classification Using Digital X-ray Mammograms

**DOI:** 10.3390/biomedicines10112971

**Published:** 2022-11-18

**Authors:** Riyadh M. Al-Tam, Aymen M. Al-Hejri, Sachin M. Narangale, Nagwan Abdel Samee, Noha F. Mahmoud, Mohammed A. Al-masni, Mugahed A. Al-antari

**Affiliations:** 1School of Computational Sciences, Swami Ramanand Teerth Marathwada University, Nanded 431606, Maharashtra, India; 2School of Media Studies, Swami Ramanand Teerth Marathwada University, Nanded 431606, Maharashtra, India; 3Department of Information Technology, College of Computer and Information Sciences, Princess Nourah bint Abdulrahman University, P.O. Box 84428, Riyadh 11671, Saudi Arabia; 4Rehabilitation Sciences Department, Health and Rehabilitation Sciences College, Princess Nourah bint Abdulrahman University, P.O. Box 84428, Riyadh 11671, Saudi Arabia; 5Department of Artificial Intelligence, College of Software and Convergence Technology, Daeyang AI Center, Sejong University, Seoul 05006, Republic of Korea

**Keywords:** breast cancer, residual convolutional neural network, Transformer Encoder (TE), self-attention mechanism, hybrid classification strategy

## Abstract

Breast cancer, which attacks the glandular epithelium of the breast, is the second most common kind of cancer in women after lung cancer, and it affects a significant number of people worldwide. Based on the advantages of Residual Convolutional Network and the Transformer Encoder with Multiple Layer Perceptron (MLP), this study proposes a novel hybrid deep learning Computer-Aided Diagnosis (CAD) system for breast lesions. While the backbone residual deep learning network is employed to create the deep features, the transformer is utilized to classify breast cancer according to the self-attention mechanism. The proposed CAD system has the capability to recognize breast cancer in two scenarios: *Scenario A* (Binary classification) and *Scenario B* (Multi-classification). Data collection and preprocessing, patch image creation and splitting, and artificial intelligence-based breast lesion identification are all components of the execution framework that are applied consistently across both cases. The effectiveness of the proposed AI model is compared against three separate deep learning models: a custom CNN, the VGG16, and the ResNet50. Two datasets, CBIS-DDSM and DDSM, are utilized to construct and test the proposed CAD system. Five-fold cross validation of the test data is used to evaluate the accuracy of the performance results. The suggested hybrid CAD system achieves encouraging evaluation results, with overall accuracies of 100% and 95.80% for binary and multiclass prediction challenges, respectively. The experimental results reveal that the proposed hybrid AI model could identify benign and malignant breast tissues significantly, which is important for radiologists to recommend further investigation of abnormal mammograms and provide the optimal treatment plan.

## 1. Introduction

Breast cancer is the second most common disease after lung cancer and affects many women worldwide because it affects the glandular epithelium of the breast [1]. This tumor is created for many reasons; however, there is no guarantee that such reasons confirm the incidence of breast cancer precisely, such as abnormal changes in breast cells or mutation in genes, etc. [2]. In 2020, with an expected 2.3 million new cases, breast cancer in females has exceeded lung cancer as the most often diagnosed malignancy, with 11.7%, followed by lung cancer, with 11.4% [3]. Therefore, many techniques have been recommended by radiologists to detect breast cancer at its curable stage, such as digital mammograms (DM), ultrasound (US), and/or magnetic resonance imaging (MRI). Generally, mammogram imaging is widely utilized for breast cancer detection at its early stage because of its low cost, superior outcomes, and convenience of operation that it provides [1,2]. By using a computer-aided diagnosis (CAD) system and mammogram images, some information regarding breast density, breast shape, and suspected abnormalities such as calcification and masses can be discovered, which helps to detect breast cancer early [1,2,4]. The mammography technique has been reported as the best modality for diagnosing the early stage of breast cancer [5]. It is not a straightforward job to distinguish cancer cell tissues from breast tissues at the early stage of breast cancer, especially for women who have dense breasts [1,2]. For example, Figure 1 shows breast mammograms for six individual cases. Both benign and malignant cases are taken from the Curated Breast Imaging Subset of the Digital Database for Screening Mammography (CBIS-DDSM) dataset, while the normal case was taken from the Database for Screening Mammography (DDSM) dataset. 

However, mammography modalities issue 2D/3D medical images with varied densities, and the available standard datasets are few. Furthermore, collecting a large dataset of mammogram images from hospitals requires hardy permission and takes a lot of time. On the other hand, some breast cases have high-intensity tissues and various features, making it difficult for radiologists to distinguish normal from abnormal tissues with the naked eye. In such cases, false-positive rates will increase, leading to the misclassification of breast images.

Meanwhile, the prediction based on the radiomic feature (i.e., engineering statistical features) is another trend to build reliable CAD systems due to their interpretability benefits of lesion characterization as well as the prediction of neoadjuvant treatment response [6,7,8]. In our previous work [4], the first-order and higher-order radiomic features were employed to build a CAD-based deep belief network (DBN) based on breast mammograms. It was clearly shown that such features were useful and trustful to provide promising performance for disease prediction. Some interesting studies as in [6,7,8] shown the importance of the relationship between imaging radiomic features with the risk of recurrence, prognosis, and molecular phenotypes using MRI imaging modality, which helps to predict malignancy probability in the breast images. Such an interesting issue could be considered for further investigation in order to build a reliable hybrid CAD system based on the deep learning image features and radiomic features altogether. This is important to detect the type of the current disease and also consider the future risk of recurrence or prognosis.

The contributions and objectives of this work are summarized as follows:A new hybrid ResNet with Transformer Encoder (TE) framework is proposed to automatically predict breast cancer from the X-ray mammographic datasets. The deep learning ResNet is used as a backbone network for deep feature extraction, while TE with multilayer perceptron (MLP) is used to classify breast cancer.A comprehensive computer-aided diagnosis (CAD) system is proposed to classify the breast cancer in two scenarios: binary classification (normal vs. abnormal) and multiple classification (Normal vs. Benign vs. Malignant).Three AI models of custom CNN, VGG16, and ResNet50 were used for performance comparison study with the proposed AI model in both binary and multi-class classification scenarios.An adaptive and automatic image processing segmentation triangle algorithm is proposed to create the adapted threshold for extracting Regions of Interest (ROIs) from the entire mammograms. The proposed algorithm leads a better segmented boundary region when compared to the conventional binary threshold segmentation.The augmentation processing is applied to increase the number of patches of the images to overcome the overfitting problems and create a large dataset that is enough for training and testing the proposed models.Four abnormal datasets were created with different patch sizes: 256 × 256, 400 × 400, and 512 × 512. The proposed models recorded the best results when using a larger patch size.

The rest of this paper is organized as follows: Section 2 summarizes the related works. Section 3 introduces the research methodology and materials used for breast cancer detection. Moreover, Section 3 presents the evaluation matrices used in this paper. The experimental results are presented and discussed in Section 4. Finally, in Section 5, the conclusion and future work are included.

## 2. Related Works

### 2.1. Deep Learning Classification 

In recent years, the medical deep learning applications of breast cancer have garnered a great deal of attention in the fields of lesions segmentation, detection, and classification [9]. Several deep learning models have been explored and developed to enhance the diagnosis rate of breast cancer based on the binary or multi-classification situation. A collection of investigations was undertaken to identify benign from malignant patients using mammography images. The deep learning YOLO predictor was applied to distinguish benign from malignant cases on mammogram images, as in the previous studies [9,10,11,12,13]. Al-Antari et al. [9] proposed a deep learning recognition framework based on the YOLO predictor for breast images detection and classification to distinguish benign from malignant cases. The YOLO was mainly used to detect the breast tumor from the entire mammograms, while for classification purpose, three classifiers were applied, namely, Regular feedforward CNN, ResNet-50, and InceptionResNet-V2. The InceptionResNet-V2 classifier achieved the best performance, among other classifiers, achieving the accuracy of 97.50% for the DDSM dataset and 95.32% for the INbreast dataset. However, the authors focused on detecting only the breast mass rather than the micro-calcification problem, since it is a different phenomenon and needs different detection techniques [14]. In another study, Hamed et al. [10] used the YOLO classifier to recognize benign from malignant breast images by proposing three processes for detecting and classifying breast cancer. They achieved the overall classification performance of 89.5% of the overall accuracy. Hamed et al. [11] utilized the YOLOV4-based CAD system to recognize the benign cases from malignant ones too, while different feature extractors such as Inception, ResNet, VGG were employed to classify the localized lesions to benign or malignant cases. The proposed model based on YOLO-V4 model outperformed other classifiers and achieved an accuracy of 98% for detecting the location of masses, while the best classification accuracy achieved by the ResNet was 95%. Aly et al. [12] used the YOLOv3 classifier to detect benign and cancer masses, while ResNet and Inception models were implemented to extract the important features. The proposed model was able to detect 89.4% of the masses, where 94.2% and 84.6% of precisions were reached for recognizing benign and malignant masses, respectively. Although the YOLO detector works efficiently to make predictions of input images, it can be difficult to detect small clustering of micro-calcification objects [15].

On the other hand, the CNN technique was used for breast cancer detection and classification as well [16,17,18,19,20,21,22,23,24,25]. Kooi et al. [16] compared a CADe system based on CNN against the traditional CADe system that relied on hand-crafted image characteristics. The final results presented that the CNN-based CADe system exceeded the traditional CADe system at low sensitivity, while the same result had provided for both suggested systems at high sensitivity. The AUC for CNN-based CADe was 0.929, while the AUC for the reference CADe system was 0.91. At the same trend, Xi et al. [17] proposed CNN models for categorizing and localizing calcifications and masses in mammography images using computer-aided detection. VGGNet had the best overall classification accuracy, according to the authors’ findings, with a score of 92.53%. Hou et al. [18] presented a study to detect calcification in mammography using a deep convolutional autoencoder based on a one-class and semi-supervised model. The proposed model reached 0.959% of AUROC and 0.676% of AUPRC during the validation stage. This model detected calcification lesions with a sensitivity of 75% and a false positive rate of 2.5% per image. According to the authors’ findings, a more advanced model or a larger dataset did not increase detection performance. In [19], an extraction method of image texture attribute and a CNN classifier were applied to develop a system for autonomously identifying breast cancer. Uniform Manifold Approximation and Projection (UMAP) was applied to minimized the extracted features. Such a model was able to distinguish normal from abnormal images and achieved 97.8% and 98% of specificity and accuracy, respectively, on the collected images from the Mammographic Image Analysis Society (MIAS) dataset, while on the images of the DDSM dataset, the model reached 98.3% and 97.9% of specificity and accuracy, respectively. Furthermore, Pillai et al. [20] used a VGG16 deep learning model to diagnose breast cancer in mammography. This model outperformed the AlexNet, EfficientNet, and GoogleNet models, with an accuracy of 75.46%. Mahmood et al. [21] developed a novel deep learning-based convolutional neural network (ConvNet) to drastically reduce human error in diagnosing breast cancer tissues. In breast masses classification, the proposed model achieved 0.98% of training accuracy, 0.97% of test accuracy, 0.99 of sensitivity, and 0.99 of AUC. Another study applied CNN model for features extraction and the Support Vector Machine (SVM) for classification stage [22]. The authors employed a fusion of various deep features step and Principal Component Analysis (PCA). Two datasets were used, MIAS and INbreast, and the proposed model achieved 97.93% and 96.646% of classification accuracy for both datasets, respectively. When PCA was applied, the computational cost and execution time were reduced; however, the classification performance was not improved. Gaona and Lakshminarayanan [23] used a CNN model that was carried out by using DenseNet architecture for detecting, segmenting, and classifying breast tumors in mammography images. The performance matrices achieved by this work were 99% of sensitivity, 94% of specificity, 97% of AUC, and 97.7% of accuracy. Shen et al. [24] investigated a group of deep learning algorithms for breast cancer detection on mammogram images of the CBIS-DDSM dataset by involving single and four models. The best single model produced an AUC per image of 88% for an independent test, while four-model averaging increased the AUC to 91%, specificity was 80.1%, and the sensitivity was 86.1%. Moreover, another dataset, INbreast, was also used, where the best single model produced an AUC of 95% per image for an independent test, while four-model averaging increased the AUC to 98%, the sensitivity to 86.7%, and specificity to 96.1%. In contrast, to avoid any degradation in image quality during the early stages, Roy et al. [25] utilized a convolution neural network (CNN) and connected component analysis (CCA) for malignant breast segmentation without any pre-processing. K-means (KM) and Fuzzy c-means (FCM) were applied for segmenting the collected images. The best achieved accuracy was 90% by using the suggested hybrid approach. Finally, the authors in [26] proposed a framework based on AlexNet, VGG, and GoogleNet for extracting the essential features of the mammograms on the INbreast dataset by utilizing a univariate technique to lower the dimensionality of the extracted features. The proposed model reached 98.98% of precision, 98.99% of specificity, 98.06% of sensitivity, and 98.50% of accuracy. 

### 2.2. Vision Transformer for Image Classification 

In another trend, the vision transformer (ViT) principle was used as a categorization system by dividing an image into fixed-size patches, to be linearly concatenated as a vector sequence for processing in a conventional converter encoder [27]. Recently, a set of researchers used such a technique to recognize benign from malignant cases, i.e., Gheflati et al. examined the performance of pure and hybrid pre-trained vision transformer models based on two breast ultrasound datasets [28], demonstrating the importance of involving the Vision Transformer technique for automatically detecting breast masses in ultrasonography. Another work used a CNN module to extract local features while a ViT module was employed to identify the global features among several areas and improve the relevant local features [29]. The hybrid model achieved a high precision of 90.77%, recall of 90.73%, specificity of 85.58%, and F1 score of 90.73%. The authors in [30] suggested a ViT-based semi-supervised learning model utilizing ultrasound and histopathology datasets, which produced superior results than CNN baseline models (VGG19, ResNet101, DenseNet201). The proposed model achieved a high precision of 96.29%, while the f1-score was 96.15%, and the accuracy reached 95.29%.

## 3. Material and Methods

In this study, a new hybrid computer-aided diagnosis is proposed based on the residual convolutional network as well as the transformer encoder (TE) with multilayer perceptron (MLP). The residual convolutional network is used as a backbone network for deep features generation, while the TE is used for the classification purpose based on the Self-attention mechanism. The proposed deep learning model involves a few steps to be fulfilled in order to improve the accuracy of breast cancer detection in mammogram images, as shown in Figure 2. Firstly, the collected medical images were in DICOM format; for simplicity, such images were converted to TIFF by using in-house MATLAB (Mathworks Inc., Boston, MA, USA) code. After that, preprocessing step was applied to remove unwanted artifacts and enhance the boundary of the segmented breast images. Then, labeling, patch images, and augmentation processes were performed. Finally, the proposed AI model was trained and tested using the generated patch images.

### 3.1. Data Acquisition and Image Collection

In this study, two standard datasets of breast images were used for developing and evaluating the proposed deep learning CAD system. The digital database of screening mammography (DDSM) [31] and the Curated Breast Imaging Subset of DDSM (CBIS-DDSM) datasets [32] were used for this work. The CBIS-DDSM dataset was revised by radiologists; therefore, some wrong or suspected diagnosed images from DDSM were removed in CBIS-DDSM, which makes this dataset suitable to be used for benign or malignant images. Both datasets have cranio-caudal (CC) and mediolateral oblique (MLO) views for left and right breast images, which means that each case (i.e., patient) has four views: two MLO and two CC views for left and right breasts sequentially. In this study, each view is treated as a separate image with its corresponding label. 

The CBIS-DDSM dataset contains 6671 breast images for 1566 patients. Indeed, CBIS-DDSM dataset is a modified and standardized version of the DDSM original dataset to only holds abnormal cases (i.e., benign and malignant), while the original DDSM dataset contains 2620 scanned film mammography images including normal, benign, malignant cases. In this work, the final created dataset has a total of 4091 breast images including normal cases collected from DDSM and abnormal cases collected from CBIS-DDSM. The total number of images per class is defined to be 998 normal, 461 benign, and 431 malignant images. We randomly split the generated dataset into 80% for training, 10% for validation, and 10% for testing. The splits are stratified to ensure that the training, validation, and test groups have the same proportion of each class. The different MLO and CC views from the same patients are preserved in the same training, validation, or testing set to avoid any accuracy bias and build a reliable CAD system.

All mammograms in both datasets are annotated by expert radiologists as they are available publicly [31,32]. As in the literature, the domain researchers always assigned the label of the original annotated breast image into its extracted patch ROIs from the that image [4,9,33]. Therefore, the label of an image is taken from the datasets metadata and kept for the created patches. For example, if an original image has a malignant label, the extracted patch images have the same malignant label and so on. Furthermore, both datasets have some information for each patient that include breast density, left or right breast, image view, abnormality, abnormality type, calc type, calc distribution, assessment, pathology, and subtlety information [31,32,34].

### 3.2. Data Preparation and Preprocessing

Mammograms images are in a DICOM format. Before using mammograms images, we used MIRCODICOM software [35] to convert all the collected images into TIFF format. TIFF format was chosen because of its ability to save DICOM images with lossless formats and be used for archiving purposes with high quality [36,37]. After converting DICOM images to TIFF format, a pre-processing step was carried out. In the pre-processing step, unwanted artifacts features were removed, and then the boundary of the breast image was smoothed. For removing unwanted artifacts from each mammogram, the image thresholding technique was used [38,39], which is an image processing segmentation technique to separate the breast X-ray attenuation pixels from the background ones. Indeed, such a process was used to convert the color or grayscale image into a binary format to easily distinguish different regions in the entire mammogram. Each image pixel value is compared to a threshold value during the thresholding process; if the pixel value is less than the threshold value, it changes to be 0 (black region or background); otherwise, it changes to the maximum value (white region). The thresholding technique has different operation types such as THRESH_BINARY, THRESH_OTSU, and THRESH_TRIANGLE that can be provided by the OpenCV library [38,39]. THRESH_BINARY is a type of simple thresholding technique that depends on a defined adaptive threshold to segment the image accordingly, while THRESH_OTSU is based on the Ostu’s thresholding technique, where the threshold value is automatically calculated for segmenting the breast and background regions [38,39]. A comprehensive investigation study was conducted in order to achieve our goal to eliminate the unwanted information from the mammograms. Experimentally, we found that the THRESH_TRIANGLE was the best [38,39,40]. The triangle algorithm examines the histogram’s shape, for example, looking for valleys, peaks, and other histogram shape aspects [40]. This algorithm depends on three steps to be carried out. First, a line between the histogram’s largest value, bmax and the lowest value, bmin on the grey level axis was defined and drawn. Second, the perpendicular Euclidean distance d between the drawn line and all points in the histogram between bmax and bmin was estimated. Finally, the threshold value was chosen based on the maximal distance between the histogram and the defined line, and a binary or mask image was generated, as shown in Figure 3b–d. The THRESH_TRIANGLE operator outperformed other operators in terms of separating the breast image from the black background, as shown in Figure 3d. The THRESH_TRIANGLE operator takes a bigger area than THRESH_BINARY, but when the masked image was smoothed, a good segmentation area will be taken without losing any part from the boundary of the breast images. After obtaining the mask of the breast image, the boundary of the breast was smoothed by using the morphological image processing technique via “morphologyEx” function to smooth and remove the noise of the mammograms [38]. Selecting the largest object was performed using the connected component analysis (CCA) technique via the function of “connectedComponentsWithStats” function [38], which can typically obtain more detailed filtering of the blobs in a binary image using connected component labeling. Finally, to construct the ROI of breast image without undesirable artifacts, the “bitwise and” function was applied to multiply the original image with its associated final binary mask image, as illustrated in Figure 3e.

### 3.3. Patch Creation

For a more accurate learning process, the deep learning model was trained based on the regions of breast lesions instead of using the whole mammograms. As it is known that the breast image size is so large compared with the breast tumor size, the weight fine-tuning process during the training time must thus focus only on the tumor regions to derive more accurate deep learning parameters (i.e., network weights and biases) [33]. In the previous works and with the lack of such an accurate patch-based CBIS-DDSM dataset [9,13,33], the prior breast lesion detection procedure was performed to automatically extract the breast lesions from the input whole mammograms. However, at this time, two procedures of patch extraction and augmentation were implemented to create the patch images from the whole breast mammograms. The first procedure was performed to collect the normal patches from the DDSM dataset, while the second approach was used to extract the abnormal benign and malignant patches from the CBIS-DDSM dataset. 

For normal patch extraction, the following producer steps were sequentially applied,
**Step 1:** After applying the data preprocessing for each image collected from DDSM, the final segmented image became ready for creating a group of tiles. **Step 2:** A set of 256 × 256 tiles or patches was created from each image. The upper threshold, lower threshold, mean, and variance of the created tiles were calculated to ensure that such tiles have part of the segmented image and not mostly empty spaces.

For abnormal benign and malignant patches extraction, the following producer steps were sequentially applied,
**Step 1:** After applying data preprocessing for each image collected from CBIS-DDSM, the segmented image was ready for creating a group of tiles. **Step 2:** The original mammograms in the CBIS-DDSM dataset have cropped patches images for benign and malignant masses that were reviewed by radiologists. So, we used them directly to create slices of 512 × 512. We used 512 × 512 because the size of the cropped patches images is different from one to another, wherein some of them are smaller than this in size, while others are bigger.**Step 3:** If the size of the cropped patch is less than 512 × 512 pixels, we put this patch in a slice of 512 × 512 starting from (0,0), and then the zero padding procedure was automatically applied to maintain the desired fixed size of 512 × 512 pixels.**Step 4:** If the cropped patch is greater than 512 × 512 pixels, more than one slice was created starting from the left to right (horizontal direction) and from up to down (vertical direction). We performed this procedure to avoid any down-sampling for the generated abnormal patches.**Step 5:** Each slice was split into two 256 × 256 tiles. 

After applying the two procedures, a total of 15,790 patch images were created including 8860 normal patches, and 6930 patches, where 3348 patches were malignant and 3582 patches were benign. This dataset was used to train and test the proposed deep learning CAD system. All normal patches were stored in a folder, and the abnormal patches were also stored in two other folders: one for benign and the second for malignant. All the names of the generated patches files have the ‘PatientID_View_Side_Tile_Tile-Number.tif’ format for normal files, while for abnormal files, the ‘PatientID_View_Side_Cropped_Cropped-Number.tif’ format was used.

### 3.4. Data Splitting

For abnormal cases, all patients who have breast masses in the CBIS-DDSM dataset were considered for this study, whereas the normal cases were collected from the DDSM dataset. Two strategies were applied to split the generated patches for Scenario A (binary classification problem) and Scenario B (multi-class classification problem). For scenario A (binary classification), the dataset was generated as reported in Table 1. 

The second strategy was followed to prepare the dataset for Scenario B (Multi-class classification). Table 2 shows the data distribution over each group: training, testing, and validation.

### 3.5. Transferring Patches

In this procedure, the two generated folders (which were named as ‘X’ and ‘Y’) hold all patches after applying the patch creation procedure. The created CSV files with patient ids were also used to guide this procedure to transfer each patch to a destination folder: train, val, or test folder. To create a dataset using the first strategy, three folders were created: train, val, and test folders, which were named ‘Tr’, ‘Va’, and ‘Te’, respectively. In each folder, two folders with ‘Normal’ and ‘Abnormal’ names were created. Therefore, to transfer all patches in ‘X’ and ‘Y’ to the subfolders of ‘Tr’, ‘Va’, and ‘Te’ folders, the file name of each patch was read; if this name existed in the patient ids list in Table 1, this patch was copied by using the name of row (‘Training’ name means to transfer a file to ‘Tr’ folder) and the name of the column to transfer this file to the correct subfolder (‘Normal’, ‘Abnormal’). The first dataset was created without applying the augmentation procedure, as shown in Table 3.

### 3.6. Data Augmentation

The procedure of data augmentation was only applied on the training set after splitting datasets based on the patient level, as mentioned the section above. The augmentation procedure is very important to create balanced datasets and remove overfitting during the training and testing process. The label of augmented patches must be kept as in the original patch image. This procedure depends on the following steps to create two datasets:**Step 1:** In order to enlarge the number of training set, new patches must be created by applying flip based on a NumPy function called ‘flip’. Two flips were applied vertically and horizontally [41] based on two returned values (1: perform flipping; 0: cancel flipping) from a NumPy function called binomial, which is responsible for drawing samples based on Binomial distribution [42]. The binomial distribution (BD) is calculated by,
(1)$$BD=\left(\begin{array}{c} n\\ N \end{array}\right)\;p^{n}\;{\left(1-p\right)}^{n-N}$$
where *n* is the number of trials, p represents the probability of success, and *N* denotes the number of successes. For this work, *n* = 1, *p* = 0.5, and *N* = 1 are experimentally optimized and used. **Step 2:** After that, a rotation was applied around the origin using different angles: [5°, 10°, 15°, 20°].

The second dataset (Dataset 2) was augmented to create a balanced dataset on the training set. In Table 3, the number of normal patches is 7116, while there are 2646 malignant patches and 2865 benign patches. Therefore, for increasing these numbers to be balanced with the normal patches, a new procedure was applied. First, for increasing the benign cases with extra patches, this procedure depends on augmenting the original patches for creating new similar ones for each patch in the benign folder starting from the first patch file to the last one. This step was repeated few times until the total number of files in the benign folder reached 7116. For each patch during the augmenting process, step 1 and the angles [5°, 10°, 15°, 20°] of step 2 in the augmentation procedure were applied, wherein the first loop 5° angle was applied to each patch, and in the second loop, 10° angle was used, and so on. The same procedure was applied for creating extra patches for malignant cases. Table 4 shows the final dataset after applying the augmentation process. In this paper, the augmentation was applied only for the training set after data splitting to avoid any overlapping bias due to the augmented instances sharing among the training, validation, and testing sets [9]. 

We noticed that 256 × 256 patch images were not enough to distinguish benign from malignant cases since some important features are divided among patches. Therefore, the patch creation was modified to create new patch images (400 × 400 and 512 × 512) without applying augmentation in which larger sizes of benign and malignant features will be taken, the difference here is that the ROI of suspected areas were divided into 400 × 400 or 512 × 512 slices without splitting each one into smaller tiles. Table 5 summarizes the newly created datasets when using 400 × 400 and 512 × 512 patch sizes. The new procedure enhances the overall accuracy, especially with 512 × 512 patches size. 

A small set of patches was removed from the generated datasets, since they reduce the overall accuracy because of their bad resolution as in Figure 4a or have white shapes that mislead the correct classification, as shown in Figure 4b,c. To exclude such patches, we asked two radiologists or clinicians to verify the patch spatial quality, sufficient information for classifications, or including some other sharpness objects such as pins or metal clothes buttons. 

### 3.7. The Suggested Deep Learning Models

In this paper, three main deep learning architectures were applied to find the best performance among them. First, a deep learning model was implemented from scratch by using an improved deep Convolutional Neural Network (CNN) model, while the second model depended mainly on the pre-trained VGG16 based on the transfer learning principle. Furthermore, the pre-trained ResNet50 model was utilized based on the transfer learning too. Finally, the Vision Transformer (ViT) was applied based on these models, so some experiments were conducted based on the three models alone, and other experiments were performed by using the ViT technique with the ResNet50 models.

### 3.8. The Custom CNN Model 

The modified model in this paper was implemented by using an improved deep Convolutional Neural Network (CNN) model that is trained on a group of patches of the mammograms to classify them into normal or abnormal cases. In this model, TensorFlow 2.7.0 was used to build a sequential model by using Keras and Scikit-Learn libraries in Python, where Keras is one of the most popular libraries used for deep learning because of its simplicity of implementing and using, while the scikit-learn library is also one of the most popular libraries for general machine learning.

The final custom CNN model comprises a VGG (Visual Geometry Group) sequential type structure with five blocks, each block containing three convolutional layers with small 3 × 3 filters, a max-pooling layer, and finally a dropout layer. After each layer in this model, batch normalization was applied, which is considered to have a regularization effect and speed up convergence. The filter applied to each convolutional layer was 3 × 3, the activation function is ReLU, and the ‘same’ is used for padding, while ‘he_uniform’ is for kernel initializer, which guarantees that the output feature maps have the same width and height. The stride and padding values implemented in this model are 1 and 0 in all layers, respectively, while the dropout rate is 25%. A stochastic gradient descent method was used, namely, Adam optimizer, which is a method that depends on adaptive estimation for the first-order and second-order moment [43]. The architecture of the custom CNN architecture used in this work is depicted in Figure 5 and Table 6, showing the output shape and the total parameters of each layer. The last layer has two units when applying binary classification and three units when multi-classification is implemented.

### 3.9. AI-Based VGG16 Model

VGG16 is a CNN that is widely involved in the classification tasks in computer vision and machine learning areas. The pre-trained VGG16 model based on the ImageNet dataset is implemented, but the classification layers were removed from this model. Therefore, two new classification layers were included for binary and multiple classifications that reached the best performance. For binary classification, the classification layer of this model had two blocks: each block consists of a conventional layer with 512 neurons, and then Batch Normalization and dropout layers with a 50% dropout rate were added, respectively. Finally, the classification layers in the multiple classifications had three blocks: each one has a conventional layer with 4090 neurons, and then Batch Normalization and dropout layers with 50% of the dropout rate were added, respectively.

### 3.10. AI-Based ResNet50 Model

ResNet-50 is a 50-layer deep convolutional neural network and has been applied for image recognition tasks. Like the pre-trained VGG16 model, the ResNet50 model was trained based on the ImageNet dataset, and the classification layers were deleted too in this paper; as a result, two blocks for binary classification and three blocks for multiple classifications were added to this model using the same configuration that was applied in the VGG16 model. The VGG16 contains 138 million parameters, while the ResNet50 has 25.5 million parameters, as well as the configuration applied in the ResNet50, which made it faster in running.

### 3.11. The Proposed Hybrid AI-Based ResNet and Transformer Encoder

A transformer is a technique based on deep learning that employs the self-attention process to apply different weights for determining the importance of each input data in an encoder-decoder format. In this paper, the vision transformer (ViT), namely, ViT-b16, was adopted based on encoder approach that was initialized with ImageNet-1K + ImageNet-21k weights [27,44,45]. The ViT-b16 model linearly combines 16 × 16 2D patches of the input image into 1D vectors to be fed into a transformer encoder that is composed of multi-head self-attention (MSA) and multi-layer perceptron (MLP) blocks, as shown in Figure 2. The MSA is used to find the relation between each patch and all other patches in a single input sequence and it employs scaled dot-product attention that can be calculated by Equation (2):(2)$$\mathrm{Attention}\left(Q,K,V\right)=Softmax\;\left(\frac{Qk^{t}}{\sqrt{d_{k}}}\;\right)v$$
where *Q* denotes to query vector, K refers to the key vector, and V is a value dimensional vector. The $d_{k}$ denotes to the variance of the product $Qk^{t}$, which has a zero mean. Moreover, the product can be normalized by dividing it by the standard deviation $\sqrt{d_{k}}$. The scaled dot-product is converted into an attention score by the SoftMax function. This mechanism is the transformer’s essential module for providing parallel attention to understanding the global content of the input image. The multi-head attention enables the model to react to input from many representation subspaces at different locations at the same time. The multi-head attention uses various, learned linear projections to linearly extend the queries, keys, and values h times and can be stated by Equation (3).
(3)$$\mathrm{MultiHead}\left(Q,K,V\right)=\mathrm{Concat}\left({\mathrm{head}}_{1},\ldots ,{\mathrm{head}}_{h}\right)W^{o}\;\mathrm{where}\;\mathrm{headi}=\mathrm{Attention}\left({\mathrm{QW}}_{i}^{Q},{\mathrm{KW}}_{i}^{K},{\mathrm{VW}}_{i}^{V}\right)$$
where the projections are parameter matrices $W_{i}^{Q}\in R^{d_{model}\;x\;d_{k}},{\;W}_{i}^{K}\in R^{d_{model}\;x\;d_{k}}$, $W_{i}^{V}\in R^{d_{model}\;x\;d_{v}}$ and $W^{o}\in R^{hd_{v}\;x\;d_{model}}$. On the other hand, the Multilayer perceptron layer (MLP) block is designed as three blocks: each one consists of a non-linear layer of Gaussian error linear unit (GELU) 40, 90 neurons, Batch Normalization, and dropout layers, where the dropping rate in all dropout layers was 50%. Furthermore, since the pre-trained ResNet50 model recorded the best performance in binary and multiple classifications compared to the VGG16 and custom CNN models, it was engaged with the ViT-b16 model, creating a hybrid model, as shown in Figure 2.

### 3.12. Evaluation Metrics

The detection and classification stages were evaluated based on the standard evaluation metrics used by many researchers, such as accuracy, recall/sensitivity, F1-score, precision, and Receiver Operating Characteristics (ROC) Curve metrics [1]. All evaluation metrics were recorded over five-fold cross-validation trails. The accuracy of a machine learning model can be calculated as a percentage or as a number between 0 and 1, and it can be performed using Equation (4). The sensitivity is a numerical value that indicates how well it can correctly diagnose patients with breast cancer and be calculated by Equation (5). The precision shows how effectively a method classifies cases correctly, and it can be accomplished by Equation (6). The F1-score is a metric that combines both precision and sensitivity in one measurement, which is calculated in Equation (7).
(4)$$\mathrm{Accuracy}\left(\mathrm{Acc}.\right)=\frac{\mathrm{TP}+\mathrm{TN}}{\mathrm{TP}+\mathrm{TN}+\mathrm{FP}+\mathrm{FN}}$$
(5)$$\mathrm{Recall}/\mathrm{Sensitivity}\left(\mathrm{SE}\right)=\frac{\mathrm{TP}}{\mathrm{TP}+\mathrm{FN}}$$
(6)$$\mathrm{Precision}\left(\mathrm{PRE}\right)=\frac{\mathrm{TP}}{\mathrm{TP}+\mathrm{FP}}$$
(7)$$F1\text{-}\mathrm{score}=\frac{2*\mathrm{Precision}*\mathrm{Sensitivity}}{\mathrm{Precision}+\mathrm{Sensitivity}}$$
where TP (True-Positive) denotes that a method accurately diagnosed the disease as positive, whereas FN (False-Negative) denotes that a method mistakenly classed the disease as negative. The term TN (True-Negative) refers to a method that appropriately classifies a disease as negative. Finally, FP (False-Positive) demonstrates that a method mistakenly classifies the disease as positive. 

Furthermore, a confusion matrix is used to assess the model classification’s performance, which is an N × N matrix where N represents the number of target classes. The confusion matrix displays both right and wrong values in the graph and reveals not only the number of errors produced by a classifier but also the sorts of errors made. For both normal and abnormal classifications, the ROC was created as a result of the tradeoff between True Positive Rate (TPR) or sensitivity and False Positive Rate (FPR) or specificity. Since the ROC curve depicts the relationship between the sensitivity and 1-specificty for the binary class problem, we used the ready built-in function of “roc_curve” from the Sklearn library, Python [46]. To obtain the ROC and its AUC value for the multi-classification scenario, we performed the strategy of one class vs. others to derive the ROC curves with their AUC values as in our previous work [4]. Then, the averaged AUC values were estimated and reported. 

### 3.13. Execution Environment

The experiment has carried out based on an ASUS laptop with the following specification: AMD Ryzen 9 5900 HX with Radeon Graphics (16CPUs), ~3.3 GHz, 32 GB of RAM, and NVIDIA GeForce RTX 3080 GPU with 16 GB. The experiments conducted in this study were implemented in Jupyter Notebook and Python 3.8.0 on Windows 11 and the TensorFlow and Keras backend libraries.

## 4. Experimental Results and Discussion

### 4.1. Scenario A: Binary Classification: Normal vs. Abnormal

For binary classification, four models were adopted and compared: the custom CNN, VGG16, ResNet50, and the hybrid (ResNet50 + ViT) models. The deep learning models of VGG16 and ResNet50 were pre-trained using ImageNet. Their pre-trained weights were used for this study based on the transfer learning strategy. All layers were non-trainable except the layer starting from 17 (‘block5_conv3′) to the output layers in the VGG16 model were trainable, while the layer starting from 143 (‘conv5_block1_1_conv’) to the last layer in the ResNet50 model were only trainable. For comparing the final results among the four models, the classifications layers on all were the same, and the optimizer used was Adam. Furthermore, different units for the output or classification layers were used, but the highest accuracies were achieved when the number of units was 512. The learning rate was 0.0001, and the number of epochs was 25 for each model. In this scenario, two types of experiments were conducted to compare the overall performance of the four models: one depends on a single test without applying the k-fold cross-validation technique, while the second experiment mainly depends on 5-fold cross-validation.

First, the hybrid model outperformed all other models by reaching 100% of overall accuracy, while the VGG16 recorded the lowest values, as presented in Table 7. Moreover, the ResNet50 wrongly predicted an abnormal patch as normal, while the VGG16 has six wrong predictions, and the custom CNN model had only four wrong predictions, and the hybrid model achieved the optimal values, based on the confusion matrices that are presented in Figure 6.

In the second experiment, for obtaining more reliable results, k-fold cross-validation was applied in the four models, which is a method of separating the dataset into train and test sets without bias [12]. The 5-fold cross-validation was conducted, which means that these models were trained five times. In this assessment, the dataset was only divided into train and test sets, where each patient’s images appeared in either train or test set in each loop of the fold cross-validation. To apply this technique, the total normal or abnormal patient numbers were divided by 5, creating five groups for both normal and abnormal patient ids, and accordingly, all patients’ images had to appear in the same group that belonged to their patient ids. For the first loop or fold, patients’ images in the first group were used for testing, and the rest groups were used for training. In the second fold, the second group was used for testing, while the remaining groups were used for training, and so on. Table 8 shows the final measure metrics for each fold test based on dataset 1.

### 4.2. Scenario B: Multi-Class Classification: Normal vs. Brnign vs. Malignant 

In this scenario, three types of experiments were conducted: the first experiment was to check the performance of the three models without using the hybrid to distinguish more difficult features (i.e., normal, benign, and malignant cases) based on datasets 2 and 3, while the second experiment was to compare the best model in the first experiment with the hybrid model by using dataset 4. The third experiment was to apply the k-fold technique to the best model (hybrid) to dataset 4. The fine-tuning applied for the ResNet50 here is different from scenario A, where it is equal to 123, since it was provided the best performance. Table 9 summarizes the final accuracies of the three models when using dataset 2 with 256 × 256 patches with 25 epochs, where the ResNet50 model outperforms other models in validation and test accuracies with values of 91%, and 90.86%, respectively.

However, even if the test and validation accuracies reach more than 90%, the precision, recall, and f1-score measures for benign and malignant were not good for all models. Therefore, new patch images were created, i.e., 400 × 400 and 512 × 512 patch images, creating datasets 3 and 4. The second experiment was conducted to check the performance of the three models without the hybrid, where the best one in the previous experiments was used as a backbone for the hybrid model. Furthermore, the number of epochs used to train models in datasets 3 and 4 was 120, while in dataset 2 it was 25, when checking the overall performance deeply. Table 10 shows the performance of the three models (VGG16, Custom CCN, and ResNet50) on dataset 3, while ResNet50 and hybrid (ResNet50 + ViT) models were only used with dataset 4, since the ResNet50 recorded the highest accuracies on datasets 2 and 3, when performing a deep investigation of the total performance. The confusion matrices were depicted for the Resnet50 model alone and hybrid models, as shown in Figure 7.

The last experiment was to apply the five-fold technique to the best model, which is the proposed hybrid AI model, and the final results are summarized in Table 11. As presented in Table 10, the proposed hybrid AI model has a reliable and feasible evaluation performance among other models.

From the generated results, the ResNet50 model outperforms other VGG16 and Custom CNN models using datasets 2 and 3. Dataset 4 was used with a larger patch size and more balanced data than datasets 2 and 3, to compare the overall accuracy between the ResNet50 alone and the hybrid (the ResNet50 + ViT) model. The transfer learning models work better when dealing with more difficult features to be distinguished as benign and malignant features than starting training with a model from scratch such as the custom CNN model. The total accuracy in all models was not optimal in scenario B; therefore, an extra experiment was conducted by applying the adaptive histogram equalization, hoping to distinguish benign from malignant features. The adaptive histogram equalization technique was used for improving the contrast of patches between suspicious lesions and normal tissues [9,47]. However, no improvement in the test and validation accuracies was noticed in the proposed models. This shows that classifying almost all benign and malignant images in the CBIS-DDSM was a challenging task. Furthermore, the regularizer technique was applied with a rate equal to 0.00005 and 0.0001 on the ResNet50 model, leading to better accuracy, especially when using 0.0001; however, when this model was used as a backbone in the hybrid model, the overfitting was increased during training stage and the overall accuracy was reduced. We noticed that when creating a bigger size of patches, the proposed models reach better results; however, it takes more time and memory to be trained. Finally, Table 12 summarizes the related studies’ results based on the DL techniques used for breast cancer detection by comparing them with the outcomes of the hybrid AI model.

Even if the hybrid model reaches the best performance in scenarios A and B, it needs more time to be trained and tested. The custom CNN model recorded good results; however, starting such a model from scratch needs a large number of images and more epochs to generalize, especially when dealing with a multiclass scenario. We noticed that a small number of epochs is sufficient for the models based on transfer learning to obtain the optimal accuracy, especially with the ResNet50, which obtains the best results after 15 epochs, while a big number of epochs increases the false-positive rate. On the contrary, new models that start from scratch need more epochs to obtain the best accuracy.

On the other hand, the study only applied a patch image classifier, so the whole image classifier will be the next intention to be implemented based on transfer learning models [24]. Furthermore, the custom CNN, VGG16, and ResNet50 models can be trained from scratch using more than one hundred thousand patches generated from CBIS-DDSM and then checking their performance by working as a patch image or whole image classifier in other datasets that have not been used before, such as INbreast or MAIS. In addition, extra transfer learning models can be analyzed including VGG19, ResNet201, and so on.

### 4.3. Limitations and Future Work

The experiments of this work for the binary classification show promising outcomes; however, the outcomes of the experiments in the multi-classification are still limited, because classifying almost all images of benign and malignant images in the CSIB-DDSM dataset is a challenging task that reduces the overall accuracy of detection. In the future, the used models and possibly other ones can be investigated on mixed images collected from datasets that have different intensities, such as INbreast, DDSM, and MAIS datasets, helping to find the best models that can deal with breast cancer images with different densities. We have a plan to continue improving the performance behavior and providing more interesting breast cancer prediction results using the newly impressive AI technologies such as explainable AI [48,49,50], federated learning [51], and so on. It is known that the medical images always have common characteristics that contain similarities in contextual features, and any deep learning model should be retuned again with respect to each modality. This is to fine tune the network trainable parameters (i.e., weights and biases). Therefore, we believe that the proposed hybrid AI model could be useful not only for breast cancer diagnosis but also for other types of tumors including liver, lung, skin cancers, and so on.

## 5. Conclusions

This research looked at how well is the proposed hybrid AI model for distinguishing abnormal (benign and malignant) from normal cases in mammography images. The evaluation was based on two publicly available databases: CBIS-DDSM and DDSM datasets. The proposed model was based on a few steps: preprocessing, segmentation and selection, and the training and testing procedures. Unwanted artifacts were removed in the preprocessing step, and the breast image’s border was smoothed. Morphological transformations were used for smoothing and removing the noise of images after selecting the ROI. Between the created mask image and the original image, OpenCV’s ‘bitwise and’ function was used to remove unwanted artifacts and to segment the required ROI of images. The generated datasets were divided into three groups: training data, test data, and validation, for binary and multiclass scenarios. The overfitting and limited data size were overcome by applying a random augmentation step by using a binomial distribution and a rotation technique to generate a set of patch images for the train set only. The hybrid model achieved a higher accuracy than the state-of-the-art studies, reaching 100% of all measure matrices for the binary classification (normal, abnormal) based on both single tests without using the cross-validation technique and the five-fold cross-validation technique. In scenario B, the hybrid model reached the best overall accuracy among others with a value of 95.8%. The patch creation was used to create three patch sizes: 256 × 256, 400 × 400, and 512 × 512. In the binary classification, 256 × 256 patches were sufficient to distinguish normal from abnormal; however, the size is not enough to recognize benign from malignant cases. In conclusion, the proposed AI hybrid model can perform detection tasks with a high degree of accuracy compared with other models. Furthermore, this model can accurately distinguish normal from abnormal patches, which can be used as a tool for differentiating normal from abnormal patches of mammograms.

## Figures and Tables

**Figure 1 biomedicines-10-02971-f001:**
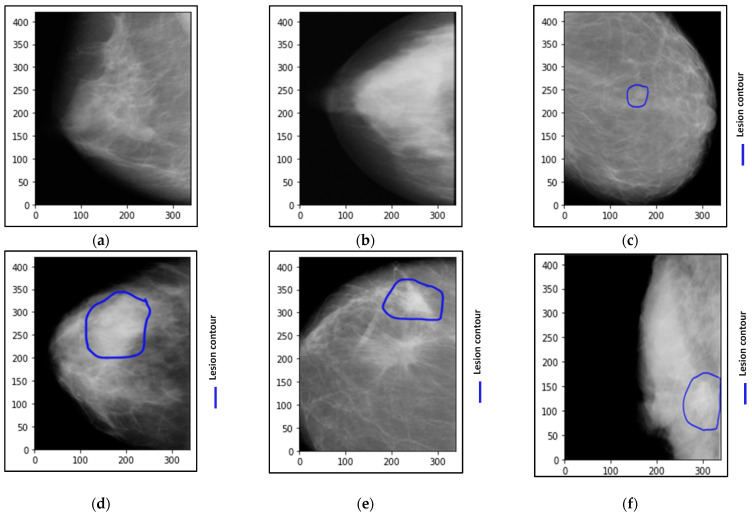
Examples of breast cancer mammograms with the abnormality localization as a blue contour around the breast lesions. (**a**,**b**) are normal cases, (**c**,**d**) are benign cases, and (**e**,**f**) are malignant cases.

**Figure 2 biomedicines-10-02971-f002:**
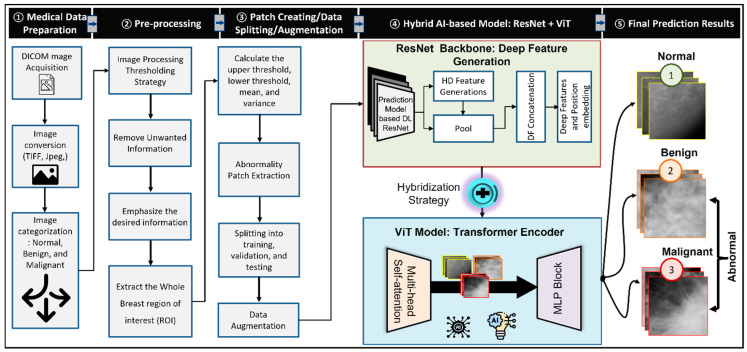
The proposed hybrid Computer-Aided Diagnosis (CAD) Model based on the ResNet backbone network and the Transformer Encoder (TE) with MLP.

**Figure 3 biomedicines-10-02971-f003:**
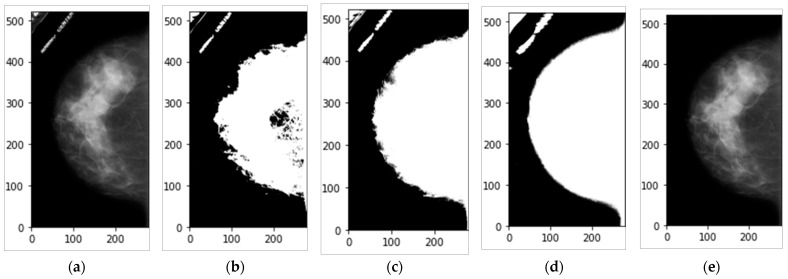
Data preprocessing to extract the whole breast region of interest (ROI) as well as remove the unwanted information using the custom built in image processing technique. (**a**) The original mammogram, (**b**) the generated image binary mask using THRESH_OTSU operator, (**c**) the generated image binary mask using THRESH_BINARY operator, (**d**) the generated image binary mask using THRESH_TRIANGLE operator, and (**e**) the corresponding breast image after applying the processing technique.

**Figure 4 biomedicines-10-02971-f004:**
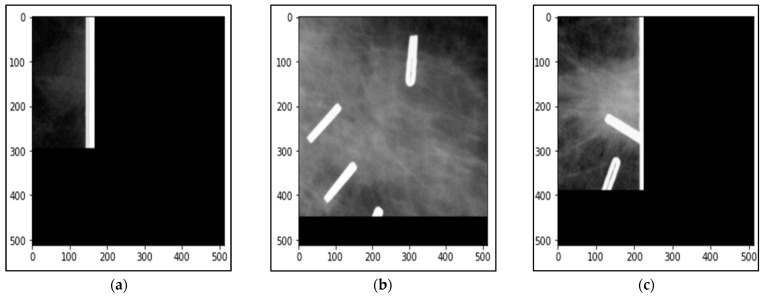
Some samples of the patches that were removed from the generated datasets. (**a**) a patch image with bad resolution, (**b**,**c**) are patches with white shapes that mislead the correct classification.

**Figure 5 biomedicines-10-02971-f005:**
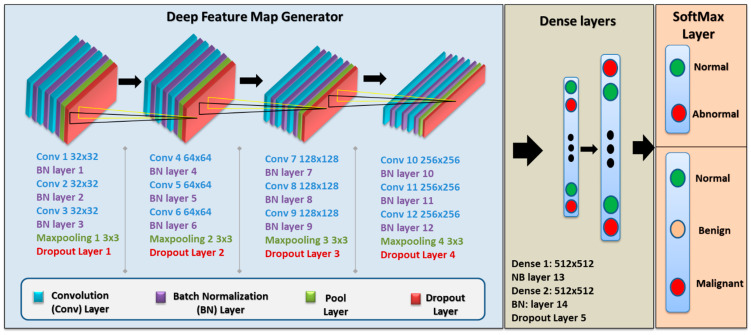
The deep learning structure of our proposed Custom CNN model.

**Figure 6 biomedicines-10-02971-f006:**
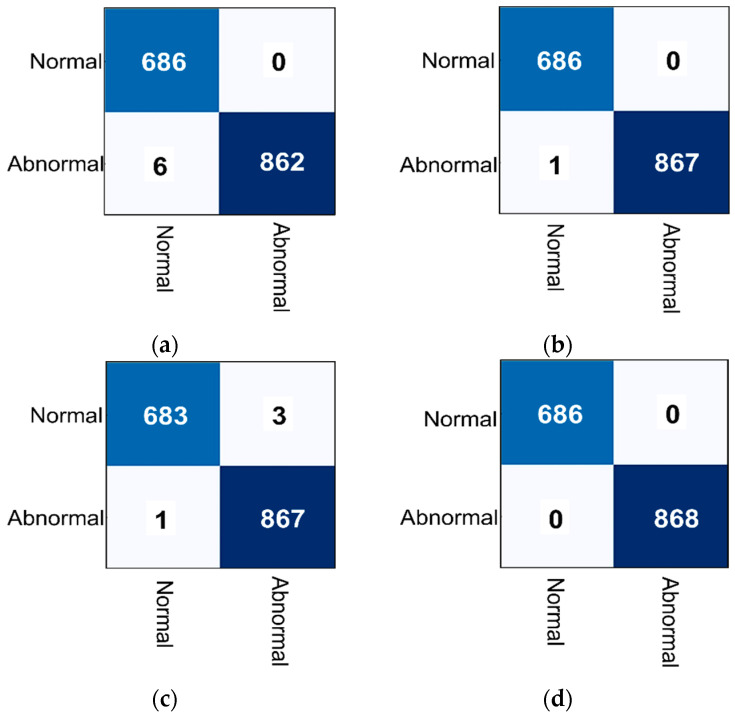
The confusion matrices of scenario A (i.e., binary classification) for the AI models: (**a**) The VGG-16, (**b**) The ResNet50, (**c**) The proposed custom **CNN** model, and (**d**) the proposed hybrid AI model.

**Figure 7 biomedicines-10-02971-f007:**
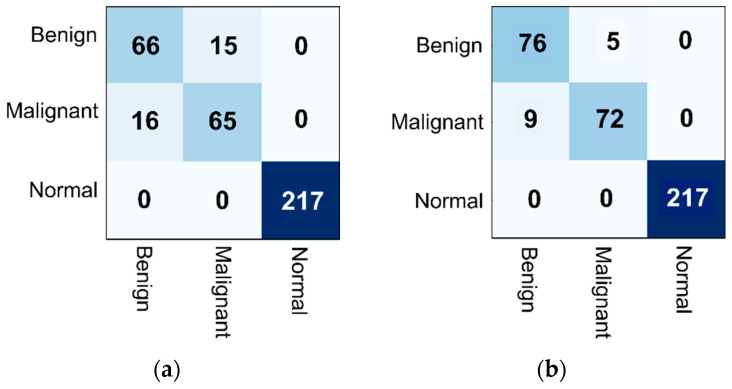
Examples of confusion matrices using (**a**) ResNet50, and (**b**) the proposed Hybrid CAD system using Dataset 4.

**Table 1 biomedicines-10-02971-t001:** Scenario A (binary Classification) data splitting training, validation, and testing.

Data Splitting	Normal	Abnormal
Training (80%)	480	713
Validation (10%)	60	90
Testing (10%)	60	89
Total Patient Number	600	892

**Table 2 biomedicines-10-02971-t002:** Scenario B (Multi-class Classification) data splitting training, validation, and testing.

Data Splitting	Normal	Benign	Malignant
Training (80%)	480	345	368
Validation (10%)	60	43	47
Testing (10%)	60	43	46
Total Patient Number	600	431	461

**Table 3 biomedicines-10-02971-t003:** Splitting based on patient level to create patches and without augmentation for normal and abnormal.

Dataset No.	Data Splitting	Normal	Benign (B)/Malignant (M)
**DataSet 1**	Training (~80.31%)	7116	5511 (B = 2865/M = 2646)
	Validation (~9.88%)	876	733 (B = 370/M = 363)
	Testing (~9.79%)	868	686 (B = 380/M = 306)
	Total	8860	6930

**Table 4 biomedicines-10-02971-t004:** Splitting based on the patient level to create (256 × 256) patches using augmentation for normal, benign, and malignant cases.

Dataset No	Data Splitting	Normal	Benign	Malignant
**DataSet 2**	Training	7116	7116	7116
	Validation	876	370	363
	Testing	868	380	306
	Total	8860	7866	7785

**Table 5 biomedicines-10-02971-t005:** Splitting based on the patient level to create (400 × 400, 512 × 512) patches using augmentation for normal, benign, and malignant cases.

Dataset No	Data Splitting	Normal	Benign	Malignant
**DataSet 3**	Training	7116	1187	1372
	Validation	876	189	177
	Testing	868	166	147
	Total	8860	1542	1696
**DataSet 4**	Training	939	895	938
	Validation	219	119	114
	Testing	217	126	98
	Total	1375	1140	1150

**Table 6 biomedicines-10-02971-t006:** The technical architecture details the proposed sequential deep learning based on CNN.

Layer	Output Shape	Parameters	Details
conv2d	(256, 256, 32)	320	3 × 3 filter, ReLU, ‘same’, ‘he_uniform’
batch_normalization	(256, 256, 32)	128	-
conv2d_1	(256, 256, 32)	9248	3 × 3 filter, ReLU, ‘same’, ‘he_uniform’
batch_normalization_1	(256, 256, 32)	128	-
conv2d_2	(256, 256, 32)	9248	3 × 3 filter, ReLU, ‘same’, ‘he_uniform’
batch_normalization_2	(256, 256, 32)	128	-
max_pooling2d	(85, 85, 32)	0	Max pooling with 3 × 3
Dropout	(85, 85, 32)	0	Dropout rate is 0.25
conv2d_3	(85, 85, 64)	18,496	3 × 3 filter, ReLU, ‘same’, ‘he_uniform’
batch_normalization_3	(85, 85, 64)	256	-
conv2d_4	(85, 85, 64)	36,928	3 × 3 filter, ReLU, ‘same’, ‘he_uniform’
batch_normalization_4	(85, 85, 64)	256	-
conv2d_5	(85, 85, 64)	36,928	3 × 3 filter, ReLU, ‘same’, ‘he_uniform’
batch_normalization_5	(85, 85, 64)	256	-
max_pooling2d_1	(28, 28, 64)	0	Max pooling with 3 × 3
dropout_1	(28, 28, 64)	0	Dropout rate is 0.25
conv2d_6	(28, 28, 128)	73,856	3 × 3 filter, ReLU, ‘same’, ‘he_uniform’
batch_normalization_6	(28, 28, 128)	512	-
conv2d_7	(28, 28, 128)	147,584	3 × 3 filter, ReLU, ‘same’, ‘he_uniform’
batch_normalization_7	(28, 28, 128)	512	-
conv2d_8	(28, 28, 128)	147,584	3 × 3 filter, ReLU, ‘same’, ‘he_uniform’
batch_normalization_8	(28, 28, 128)	512	-
max_pooling2d_2	(9, 9, 128)	0	Max pooling with 3 × 3
dropout_2	(9, 9, 128)	0	Dropout rate is 0.25
conv2d_9	(9, 9, 256)	295,168	3 × 3 filter, ReLU, ‘same’, ‘he_uniform’
batch_normalization_9	(9, 9, 256)	1024	-
conv2d_10	(9, 9, 256)	590,080	3 × 3 filter, ReLU, ‘same’, ‘he_uniform’
batch_normalization_10	(9, 9, 256)	1024	-
conv2d_11	(9, 9, 256)	590,080	3 × 3 filter, ReLU, ‘same’, ‘he_uniform’
batch_normalization_11	(9, 9, 256)	1024	-
max_pooling2d_3	(3, 3, 256)	0	Max pooling with 3 × 3
dropout_3	(3, 3, 256)	0	Dropout rate is 0.25
flatten	(2304)	0	-
dense	(512)	1,180,160	-
batch_normalization_12	(512)	2048	-
dense_1	(512)	262,656	-
batch_normalization_13	(512)	2048	-
dropout_4	(512)	0	-
dense_2	(2)	1026	-
**Total parameters**	**3,409,218**		**-**
**Trainable parameters**	**3,404,290**		**-**
**Non-trainable parameters**	**4928**		**-**

**Table 7 biomedicines-10-02971-t007:** Scenario A evaluation performance via four AI models using Dataset 1 without cross-validation.

AI Model	Fine-Tune Layers	Class	Evaluation Measurements (%)					
			PRE	SE	F1-Score	Val. Acc.	Test Acc.	AUC
**VGG16**	17	Normal	99.60	98.60	98.80	98.64	98.60	98.67
		Abnormal	98.40	98.60	98.20			
**ResNet50**	143	Normal	99.60	100.0	99.80	99.83	99.76	99.81
		Abnormal	100.0	99.60	99.80			
**Custom CNN**	Trained from scratch	Normal	99.20	99.00	99.20	99.17	99.25	99.28
		Abnormal	98.99	99.10	99.19			
**The Proposed Hybrid AI Model**	143	Normal	100.0	100.0	100.0	100.0	100.0	100.0
		Abnormal	100.0	100.0	100.0			

**Table 8 biomedicines-10-02971-t008:** Scenario A evaluation results (%) using Dataset 1 as an average over five-fold tests.

Model	Fine-Tuned Layers	Class	PRE	SE	F1-Score	Test Acc.	AUC
**VGG16**	17	Normal	99.62	98.65	98.84	98.61	98.68
		Abnormal	98.40	98.68	98.26		
**ResNet50**	143	Normal	99.69	99.81	99.89	99.83	99.89
		Abnormal	100.0	99.69	99.82		
**Custom CNN**	Trained from scratch	Normal	99.27	99.05	99.26	99.20	99.26
		Abnormal	98.84	99.12	99.30		
**The proposed hybrid AI Model**	143	Normal	100.0	100.0	100.0	100.0	100.0
		Abnormal	100.0	100.0	100.0		

**Table 9 biomedicines-10-02971-t009:** Scenario B (Multi-class classification) evaluation performance (%) via three AI models using Dataset 2 (256 × 256 patches) without cross-validation.

Models	Class	PRE	SE	F1-Score	Val. Acc.	Test Acc.
**VGG16**	Benign	60.10	65.70	62.31	87.45	87.75
	Malignant	63.04	54.00	58.20		
	Normal	99.02	100.0	99.00		
**ResNet50**	Benign	80.11	59.50	67.00	91.00	90.86
	Malignant	54.90	78.51	65.00		
	Normal	100.0	100.0	100.0		
**Custom CNN**	Benign	59.10	65.21	61.00	87.03	87.20
	Malignant	63.10	52.23	57.06		
	Normal	98.02	100.0	99.60		

**Table 10 biomedicines-10-02971-t010:** Multi-class classification evaluation performance (%) among different AI models using dataset 3 and dataset 4.

Models	Dataset	Class	PRE	Recall	F1-Score	Val. Acc.	Test Acc.
**VGG16**	Dataset 3	Benign	80.11	59.01	68.03	90.75	90.25
		Malignant	55.89	78.00	65.00		
		Normal	100.0	100.0	100.0		
**ResNet50**		Benign	87.12	69.98	78.00	94.10	93.33
		Malignant	65.89	83.12	73.02		
		Normal	100.0	100.0	100.0		
**Custom CNN**		Benign	79.10	58.00	67.10	90.00	89.97
		Malignant	53.90	76.00	62.90		
		Normal	100.0	100.0	100.0		
**ResNet50**	Dataset 4	Benign	83.22	91.80	87.00	94.02	92.19
		Malignant	91.10	80.87	86.12		
		Normal	100.0	100.0	100.0		
**The Proposed Hybrid AI Model**		Benign	93.11	85.12	89.02	96.03	95.60
		Malignant	86.00	93.55	90.11		
		Normal	100.0	100.0	100.0		

**Table 11 biomedicines-10-02971-t011:** Scenario B evaluation results using dataset 4: evaluation metrics for the proposed hybrid AI model (ResNet50 and ViT) over five-fold tests.

Model	Fine-Tuned Layers	Fold	Class	PRE	SE	F1-Score	Acc.	AUC
**The Proposed Hybrid AI Model**	123	Fold 1	Benign	93.00	85.89	89.13	96.03	96.04
			Malignant	86.10	93.95	90.20		
			Normal	100.0	100.0	100.0		
		Fold 2	Benign	86.22	95.11	90.02	95.84	96.01
			Malignant	94.40	84.04	89.12		
			Normal	100.0	100.0	100.0		
		Fold 3	Benign	85.90	91.85	89.11	95.06	95.12
			Malignant	91.00	83.97	86.98		
			Normal	100.0	100.0	100.0		
		Fold 4	Benign	85.97	99.09	92.00	96.00	96.06
			Malignant	98.70	84.92	90.94		
			Normal	100.0	100.0	100.0		
		Fold 5	Benign	89.06	94.11	92.00	96.10	96.00
			Malignant	94.00	89.00	91.12		
			Normal	100.0	100.0	100.0		
	Avg. (%)		Benign	88.03	93.21	90.45	95.80	95.84
			Malignant	92.84	87.17	89.67		
			Normal	100.0	100.0	100.0		

**Table 12 biomedicines-10-02971-t012:** Comparison evaluation study with the latest AI-based breast cancer classification models.

Reference	Dataset	Prediction Classes	Deep Learning Method	Acc. (%)
Al-antari et al. (2018) [9]	DDSM	Benign/Malignant	YOLOV2	97.50
Melekoodappattu et al. (2022) [19]	DDSM	Normal/Abnormal	CNN model	98.3 (SE) and 97.9 (Acc.)
Shen et al. (2019), [24]	CBIS-DDSM	Cancer/Normal; Background/Malignant Mass/Benign Mass/Malignant Calcification/Benign Calcification	Many CNN models	91.00 (AUC), 86.1 (SE), and 80.1 (PRE)
Roy et al. (2022), [25]	DDSM	Malignant Detection	CNN with connected component analysis (CCA)	90.00
Xi et al. (2018), [17]	CBIS-DDSM	Calcification/Mass	CNN model	92.53%.
**Our**	*** DDSM and** **CBIS-DDSM**	**Scenario A: Normal/Abnormal**	**The proposed hybrid ResNet50 and ViT model**	**100**
		**Scenario B: Normal/Benign/Malignant**		**95.80**

* All normal breast images were collected from the DDSM dataset, while the benign and malignant ones were collected from CBIS-DDSM dataset.

## Data Availability

The datasets used in this paper are publicly available at: https://wiki.cancerimagingarchive.net/pages/viewpage.action?pageId=22516629 (accessed on 15 November 2022) and http://www.eng.usf.edu/cvprg/Mammography/Database.html (accessed on 15 November 2022).

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
