# Peer review of "A Hybrid Workflow of Residual Convolutional Transformer Encoder for Breast Cancer Classification Using Digital X-ray Mammograms"

_biomedicines, 2022, doi:10.3390/biomedicines10112971_

Round 1
Reviewer 1 Report
General Comments
I believe this paper represents a novel and significant contribution to the use of artificial intelligence (AI) algorithms to interpret full-field digital mammograms. That said, it is simply far too long, at least double the length that is reasonable.
In the Introduction, beginning in line 7 on page 2, until the end of the section on page 3, the authors present their Materials and Methods, and then repeat all of this information beginning on page 6. The literature review, on pages 3 to 6, is 1700 words in length, and has hardly a paragraph break. Table 8 could be shortened considerably by simply presenting averages. I could go on, but I have made my point: this manuscript is far too long and the length detracts from the important message the authors wish to convey.
Perhaps the most important results are contained in Table 12 where the authors' results for their hybrid model are compared with other results in the literature, particularly those groups that also used the CBIS-DDSM dataset.
I recommend the inclusion of a brief listing of the acronyms employed (CNN, VGG, YOLO, etc.).
Specific Comments
page 1 line 21 ... system for breast lesions.
page 1 line 23 The proposed CAD system has ...
page 2 line 9 ... helps to detect breast cancer early.
page 6 line 2 ... for automatically detecting breast masses ...
page 10 line 14 ... on the training set after splitting ...
page 10 line 15 ... as mentioned in the section above.
page 10 line 32 ... training set.
page 12 line 1 A small set of patches was removed from ...
page 12 line 2 ... because of their poor resolution or have white shapes that mislead the ...
page 13 line 1 ... this work is depicted in ...
page 17 Table 8 Why are the models not presented in the same order as in Table 7?
page 19 line 1 Normal vs Benign vs Malignant
page 19 Table 9 Spelling of "Accuracy" is last two column headings
page 23 line 17 ... hybrid AI model for distinguishing abnormal ...
page 23 line 25 ... 256x256 patches were sufficient to distinguish ...
page 24 The Funding and Acknowledgements say the same thing.
page 24 Ref. 7 This reference, which is an important one, is incomplete
Author Response
We have answered all of the reviewer's comments in the revised manuscript and have also attached all the answers in a Word file.

Reviewer 2 Report
The authors have proposed a new hybrid deep learning model for binary and multiclass classification and breast cancer detection in mammogram images. The hybrid system utilizes a residual deep learning network as the backbone to create the deep features and the transformer to classify breast cancer according to the self-attention mechanism. The effectiveness of the proposed AI model was compared against three separate deep learning models: a custom CNN, the VGG16, and the ResNet50. Two datasets, CBIS-DDSM and DDSM, were utilized to construct and test the proposed CAD system. The presented work has the potential for publication. However, the following clarification and modifications are required accordingly before the publication:
1. In section 3.10., the sentences: “In this paper, the vision transformer (ViT), namely, VIT-b16, was adopted based on encoder approach that was initialized with ImageNet-1K+ImageNet-21k weights [26], [40], [41]. The ViT-b16 model linearly combines 16×16 2D patches of the input image into 1D vectors to be fed into a transformer encoder that is composed of multi-head self-attention (MSA) and multi-layer perceptron (MLP) blocks.” Somewhere ViT and somewhere VIT has been used as the abbreviation for vision transformer.
2. In section 4.1: Table 7: The column for validation and test accuracy has already been given; then what is the column naming:”Acc.”? Also, the validation and test accuracy are exactly the same for each network; please ensure that the different data sets have been used for validation and testing the models, as you claimed in the “Material and Methods” section.
3. In the results section, the AUC and test accuracies are the same for each network. Explain the method employed for generating the AUC and ensure the results of AUC once again.
4. A similar format should be maintained in each table of the results section. For example, in table 7, The values are in decimal form, but in the other tables, the digits after the decimal are not presented. Maintain a similar format throughout the article.
“Limitation and Future Work” should be in a separate section before the conclusion.
Author Response
We have modified all of the reviewer's comments in the revised manuscript and have also attached all the answers in a Word file.

Reviewer 3 Report
This is a contribution with interesting novelties. The rationale is clear and well stated, and the implementation choices made are well justified. The hybrid deep-learning approach proposed were is well-described from both the methodological and architectural points of view. The experimental findings obtained are encouraging. In order to have a more comprehensive and correct contribution the following comments should be approached.
Comment 1 (section 1.'Introduction')
Similar to deep-learning classifiers - such as the one proposed in this study - studies exploting radiomic features try to model many kind of clinical outcomes, as lesion characterization and prediction of neoadjuvant treatment response, with the advantage of a major features interpretability. To provide a comprehensive initial scenario, also radiomic-based studies should be considered. Some interesting radiomic studies in breast that could be added and discussed are the following:
- Fusco, R., Granata, V., Maio, F., Sansone, M., Petrillo, A. (2020). Textural radiomic features and time-intensity curve data analysis by dynamic contrast-enhanced MRI for early prediction of breast cancer therapy response: preliminary data. Eur Radiol Exp 4, 8. https://doi.org/10.1186/s41747-019-0141-2
- Wang, Q., Mao, N., Liu, M., Shi, Y., Ma, H., Dong, J., Zhang, X., Duan, S., Wang, B., & Xie, H. (2021). Radiomic analysis on magnetic resonance diffusion weighted image in distinguishing triple-negative breast cancer from other subtypes: a feasibility study. Clinical imaging, 72, 136–141. https://doi.org/10.1016/j.clinimag.2020.11.024
- Militello, C., Rundo, L., Dimarco, M., Orlando, A., Woitek, R., D'Angelo, I., Russo,G., Bartolotta, T. V. (2022). 3D DCE-MRI radiomic analysis for malignant lesion prediction in breast cancer patients. Academic Radiology, 29(6), 830-840. https://doi.org/10.1016/j.acra.2021.08.024
Comment 2 (3. Material and Methods )
Images labeling ("Then, labeling, patch images, and augmenta-tion processes were performed."have been verified by a clinician?
Comment 3 (3.2. Data Preparation and Preprocessing )
Please, provide further details concerning the modality THRESH_BINARY, THRESH_OTSU, THRESH_TRIANGLE.
Comment 4
If possible - in order to ensure the repetability of the results - I think it is essential to share the obtained code. To this aim GitHub (or equivalent platform) could be used to share che implemented codewith the community.
Author Response

(The authors gave the same response as above.)

Reviewer 4 Report
In this article, Riyadh et al. developed a hybrid workflow of residual convolutional transformer encoder for breast cancer classification using digital X-ray mammograms. The computer-aided diagnosis system achieves 100% accuracies for binary predictionm and 95.80% accuracy for multiclass prediction. I think the results are encouraging for researchers on the field of oncotherapy and the work is recommended for publication in Biomedicines after some minor revisions.
1. Scale bars should be provided with Figure 1, Figure 3, and Figure 4.
2. What about the possibility for this AI technology for diagnosis of other types of tumors, such as lung cancer, lung cancer, and liver cancer?3. It is suggested to provide Information (age, condition) about the case of breast cancer patient.
Author Response

(The authors gave the same response as above.)

Round 2
Reviewer 1 Report
General Comments
Although the manuscript has only been shortened by a single page -- from 26 to 25 pages -- I am satisfied the authors have made a good faith effort to address my concerns about length.
I have checked the author guidelines for Biomedicines, and these state that the journal "has no restriction on the maximum word count or number of Figures/Tables included in the text, provided that the text is concise and comprehensive." I therefore accept this policy about manuscript length.